

# Automated Enclosure and Protection System for Compact Solar-Tracking Spectrometers

Ludwig Heinle[1] and Jia Chen[1]

[1]Environmental Sensing and Modeling, Department of Electrical and Computer Engineering, Technische Universität München, Munich, 80333, Germany

*Correspondence to:* Jia Chen (jia.chen@tum.de)

**Abstract.** A novel automated enclosure for protecting solar-tracking atmospheric instruments was designed, built, and tested under various weather conditions. A complete automated measurement system, consisting of a compact solar-tracking Fourier Transform spectrometer (*EM27/SUN*) and the enclosure, has been deployed in central Munich for greenhouse gas monitoring for one year and withstood all critical weather conditions, including rain, storm, and snow. It has provided continuous ground-based measurements of column-averaged concentrations of $CO_2$, $CH_4$, $O_2$ and $H_2O$.

The enclosure protects the instrument from harmful environmental influences while allowing for open path measurements in sunny weather conditions. The newly developed and patented cover, a key component of the enclosure, permits unblocked solar measurements, while reliably protecting the instrument within less than 6 seconds. This enables very dynamic decisions about taking measurements, and thus increases the amount of data samples.

The presented enclosure leads to a fully automated measurement system, which collects data whenever possible without any human interaction. The functionalities of the enclosure give full control over the *EM27/SUN*. It provides the fundament for a long-term greenhouse gas monitoring sensor network.

## 1 Introduction

Anthropogenic greenhouse gas (GHG) emissions into the atmosphere have risen to a worrying level over the last decades. There is little doubt that this impacts the climate on earth and eventually the wellbeing of mankind.

The understanding of greenhouse gas sources, sinks and transportation requires reliable and precise atmospheric concentration measurements. State of the art ground-based and space-borne spectrometers are used to measure column-averaged gas concentrations by analyzing the gas absorption of specific frequencies of the sunlight.

Ground-based solar-viewing spectrometers use the sun as light source. Gas molecules such as $O_2$, $CO_2$ and $CH_4$ interact with the sunlight on its path through the atmosphere, resulting in absorption lines in the recorded sun spectrum. By observing the intensity attenuation of light in specific frequencies, the concentration of gas molecules within the examined air column can be determined.

Total carbon column observing network (TCCON) (Wunch et al. (2011); Toon et al. (2009)) is a global network measuring total column concentrations. The instruments deployed in the network are *Bruker IFS 125HR* spectrometers, which provide





accurate data to validate satellite observations. However, these instruments are very large (close to 2x3m footprint), weight more than half a ton and are extremely expensive (Bruker Optic GmbH (2006)). Further, their operation is costly in terms of manpower. Therefore, many working groups operating *125HR* spectrometers use remote control or automated systems for the operation. A team at the *Belgian Institute for Space Aeronomy* developed *BARCOS* (Neefs et al. (2007)), which stands

for "Bruker automation and remote control system". Further, Geibel et al. (2010) developed an automated system for the deployment of *125HR* in Ascension Island. The main goal of these systems is to lower the operational costs by reducing the need of local and total human interactions for the operation. However, operation of an *125HR* still requires regular skilled on-site attendance, due to the degradation of interferometric alignment given by the scanner wear on the time scale of months (Hase (2012)).

A smaller tabletop spectrometer, the *EM27/SUN* (Gisi et al. (2011, 2012); Hase et al. (2016)) has found lots of applications in urban greenhouse gas emission studies in the recent years (Hase et al. (2015); Chen et al. (2016); Franklin et al. (2016)). It is lightweight, compact, and very robust and has reached a comparable precision to TCCON instruments (Frey et al. (2015); Hedelius et al. (2016); Chen et al. (2016)). It is easy to transport and operate, and therefore a much better flexibility in site selection is permitted. While the large *IFS 125HRs* are deployed for global observations, as done by TCCON, the small

*EM27/SUN*s are more often utilized for local and city source investigations (Hase et al. (2015); Chen et al. (2016); Franklin et al. (2016); Viatte et al. (2017); Chen et al. (2017)). Further, it has been deployed on a ship (Klappenbach et al. (2015)) and in a mobile observatory to assess volcanic emissions (Butz et al. (2017)). However, up to this day a lot of human manpower is necessary to operate the instrument at the measurement site. A person needs to set up the system every time before starting measurements. Since the *EM27/SUN*s are not water proof, the measurement setup needs to be manually dismantled and stored

safely, whenever the weather changes to rainy or stormy conditions. Besides wear and tear at the connectors, a lot of costly human effort is necessary. Even though the operators do not need to be highly trained, their permanent attention is required. Consequently, a very limited willingness to measure every single hour during good conditions reduces the amount of data that is collected.

An automated enclosure (Figure 1) that allows for an easy remote operation of the measurement system has been developed

and is described in this paper. It eliminates the need to dismantle the system after every measurement period, and protects against harmful weather conditions. A remote operator can take care of multiple stations at the same time while the setup and dismantling times can be reduced to a minimum. Nevertheless, the most important function of such an enclosure is the ability to reliably protect the spectrometer and its components. The easy handling combined with the small footprint and low power consumption reduce the operational costs of such a measurement system to nearby zero.

## 2 Hardware and Setup

The *EM27/SUN* is a portable solar-tracking FTIR-Spectrometer. It is equipped with a solar-tracker unit, based on motorized rotating mirrors to direct the light into the spectrometer. An external laptop is connected to the spectrometer via multiple connections. All components are small and lightweight enough for easy transportation. Unfortunately, these components are





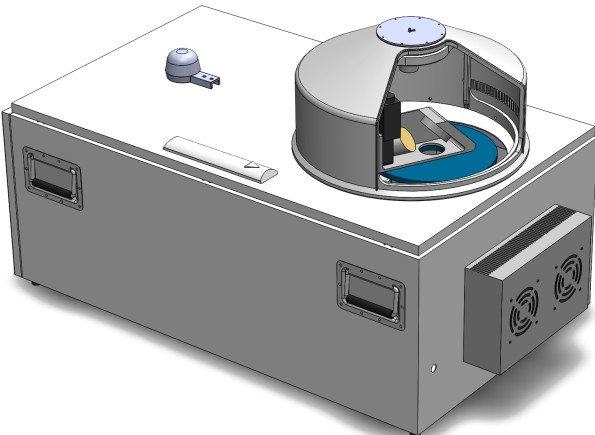

**Figure 1.** 3D Computer-Aided Design (CAD) of the enclosure including the base cabinet, the cover, the thermal electrical cooler, and the rain sensor. These hardware components will be explained in Section 2.

not weather proof, so that the system must be stored at a weather protected location. During measurements the solar-tracker's first mirror needs a direct non-obstructed line of sight to the sun. Thus, the instruments need to be repositioned to an outdoor location. Thereby, transportation, setup and observation require a lot of human effort, which can be eliminated by an automated enclosure, storing and protecting the measurement system while allowing measurements during good weather conditions.

Even though the *EM27* body itself is weather resistant, an enclosure is required for the protection of other components, e.g., the laptop, the control electronics, and the solar tracker, especially the tracker mirrors. Hedelius et al. (2016) found degradation of the solar tracker mirrors, which could be caused by a combined action of reactive substances, sea-salt aerosols, and humidity above local saturation, in which case, the contaminated water droplets might be deposited on the mirrors.

    To build such an enclosure there are several challenges to be solved. A safe and reliable protection against environmental

influences like dirt, thunderstorms and even hail is needed for the measurement system. For the measurement itself though, it is necessary that the sun tracking mirrors of the instruments can be directly exposed to the sunlight during sunny weather conditions. This leads to the prime directive for the enclosure design: reliable protection and maximized amount of measurement data.

    Other criteria include best possible remote controllability over the internet, thermal stability of the system, as well as a stable

and uninterrupted power supply. The later ensures error free collection of data and prevents the system from unpredictable states. All in all, the enclosure is aimed for automatic operation without the need of human attendance or human interactions.

    An overview of the components of the automated enclosure is given in Figure 2. These components are explained in the following subsections.




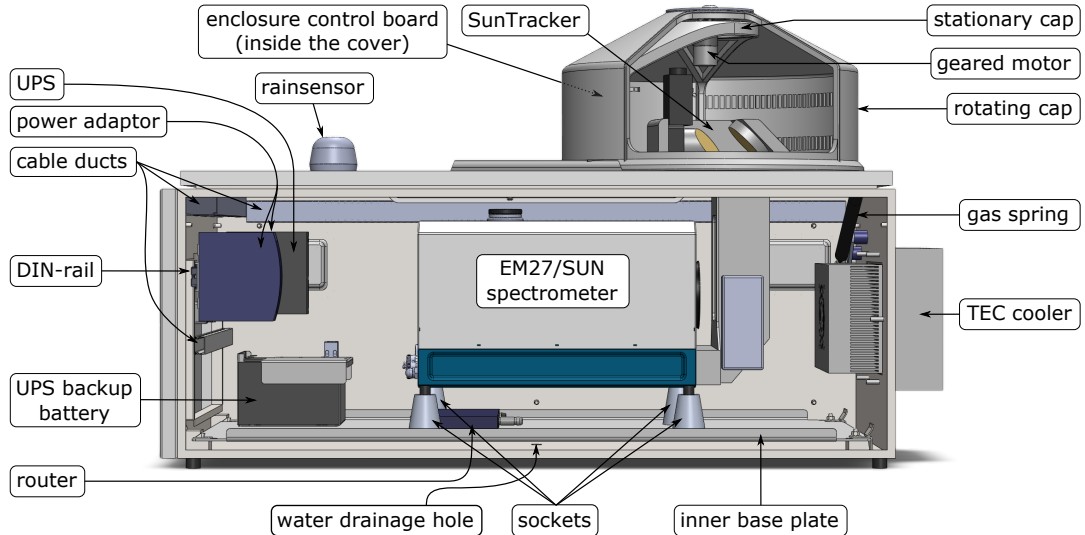

**Figure 2.** Enclosure overview: components and their arrangements.

## 2.1 Base Cabinet

As mentioned before, the *EM27/SUN* spectrometer needs to be physically protected. Therefore, a stable and waterproof control cabinet is turned on its back and serves as a base cabinet of the enclosure. In this orientation, the door is located on top and allows easy access to all equipment inside. Rubber feet at the bottom ensure a slightly elevated stable stand.

5      The electrical components of the enclosure, such as power distribution and power supplies, are well-ordered on a DIN-rail. All wirings are stowed in cable ducts to keep the inside tidily arranged, easy to access and reliable in operation.

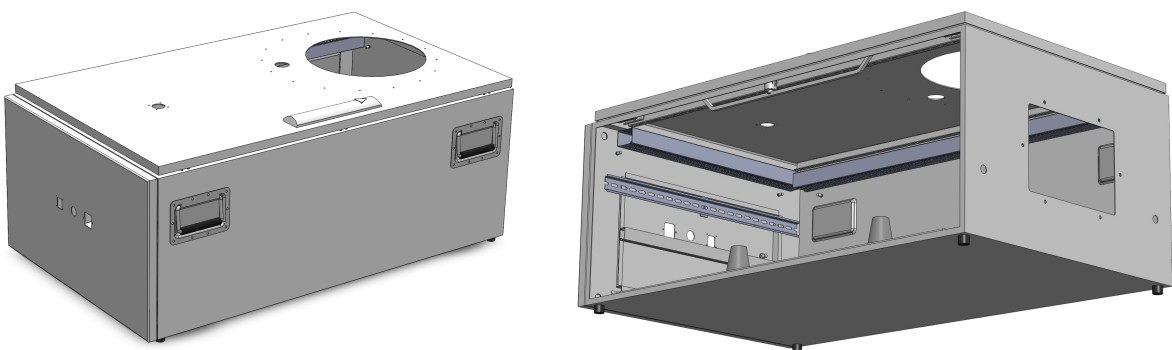

**Figure 3.** 3D-Model of the base cabinet. In the left figure the original function as a control cabinet is clearly identifiable. The handles on the side wall are mounted for easy transportation. The right image shows another perspective, with one of the side walls removed to allow a view inside the empty base cabinet.




The base cabinet, as depicted in Figure 3, is manufactured in steel. Thus, it is very strong and durable. Four handles, one on each corner, allow easy transportation. The measurement system easily fits inside and is well protected by the base cabinet. The upper end of the solar-tracker is the only part of the measurement system, that is located outside of the base cabinet. It rises through an opening in the door of the base cabinet. A cover, as described in the following Section 2.2, is mounted over the opening, to close the enclosure and protect the instrument.

## 2.2 Cover

As mentioned before, the solar-tracker extends through the top of the enclosure. Thus it needs a separate protection against harmful environmental influences. Tests with a glass dome were made at the Karlsruhe Institute of Technology (KIT) (cf. Sha (2015)). Unfortunately, the results were not as good as expected, since it was not possible to manufacture a dome with a sufficient homogeneous surface. The sunlight was distorted differently based on the solar incident angle, which disturbed the solar-tracker software. Other than that, dirt and dust on the glass surface as well as possibly trapped humidity inside the dome could disturb the measurements.

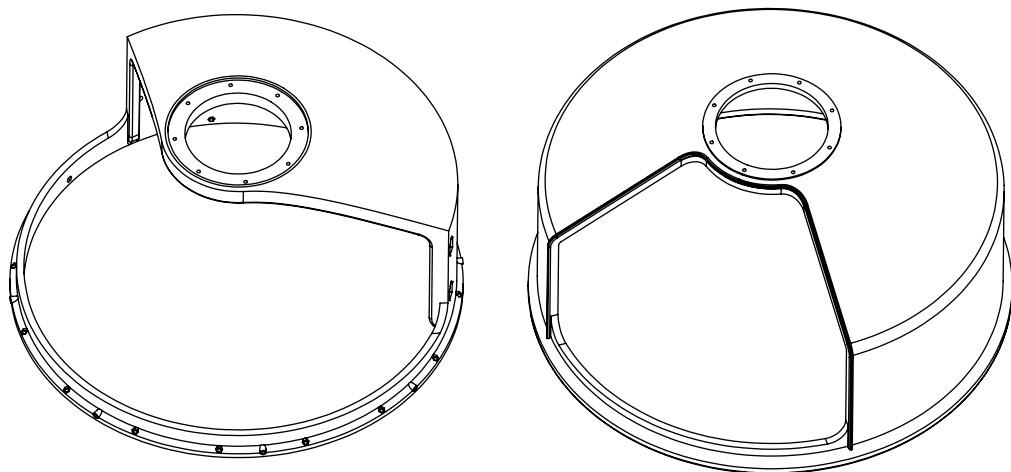

**Figure 4.** Perspective drawing of the two caps of the cover. The inner cap with its large opening is depicted on the left. The cap on the right is the outer one with a relatively small cutout.

### 2.2.1 Cover Design

A completely different concept is chosen here. A newly developed and patented cover is mounted on top of the opening, covering the solar-tracker. The cover is made of two rotationally symmetric caps with cutouts, where one cap fits inside of the other (c.f. Figure 4 and Figure 5). It protects the solar-tracker from bad weather while enabling open path measurements in dry and good weather conditions.





The outer cap is pivoted about the vertical symmetry axis on top of the inner cap. Its weight is carried by eight ball bearings that are mounted equally distributed around the lower end of the inner cap. While the inner cap will be mounted in a stationary position on top of the enclosure, the outer one can rotate. The rotation is driven by a simple geared electric motor, which is mounted from inside in the common axis of the two caps.

### 2.2.2 Size of the Cover

As depicted in Figure 4, the inner cap has a large cutout. It enables direct sunlight to hit the sun tracking mirrors whenever the solar zenith angle (SZA) is less than 80°. The most extreme azimuthal angles will happen on the longest day of the year, which covers a range of approximately 228° for a SZA below 80° in Munich. This cover design is sufficient for all places on earth with an absolute value of the latitude of less than or equal to 48°. For other places on earth closer to the poles a simple redesign of the cutouts is needed. This can be done by a simple modification of some parameters in the 3D-Model, which was made to construct the enclosure.

The opening in the outer, rotating cap (see Fig. 4) has upper and lower size limitations. The lower size limit of the opening is given by the width of the mirrors. Since the sunlight is not allowed to be blocked at sunny conditions, and the sun rays can be assumed as parallel, the opening must be at least the width of the mirrors. However, in this extreme case the cover needs to track the solar azimuth angle very precisely during the course of the day, so that it does not block the sunlight off the mirrors. Furthermore, the smaller the opening is, the accessibility inside the cover is more difficult in case of any service demands.

The upper size limit for the opening is the size of the inner cap's remaining wall. During bad weather conditions or darkness, the two caps will rotate to a closed state where the inner cover wall needs to be fully overlapped with the opening of the outer one. Sealing concerns and a lower demand of the positioning precision push the design rules towards more overlap.

Finally, an opening of 90° was considered to be a good trade off between overlapping and required tracking accuracy for the first prototype.

### 2.2.3 Weather Proof Concept

A gap between the two caps ensures a friction-free smooth rotation of the outer cap. However, rain combined with strong winds could blow water into the cover despite the overlap. Thus, a gasket as depicted in Figure 5 is added to the cover. It is made out of a window sealing strip that is cut into the right height. When the cover is closed, the gasket will block the wind and thus prevent water from being blown in. In addition, it fills the gap between the two covers and thus keeps insects from going inside. Nevertheless, the weak rubber sealing will not obstruct a smooth rotation of the outer cap.

### 2.2.4 Position Determination

Because of the 90° opening, the outer cap needs to track the sun for undisturbed measurements. Therefore, the actual position of the outer cap needs to be determined.


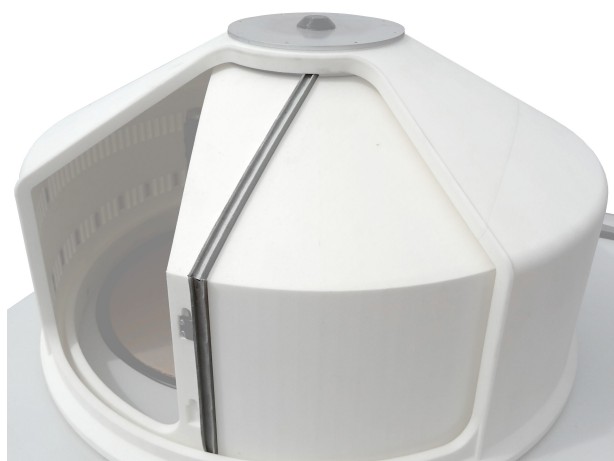

**Figure 5.** The image shows the gasket that were installed to seal the gap between the two caps. It prevents water from being blown into the cover by strong winds.

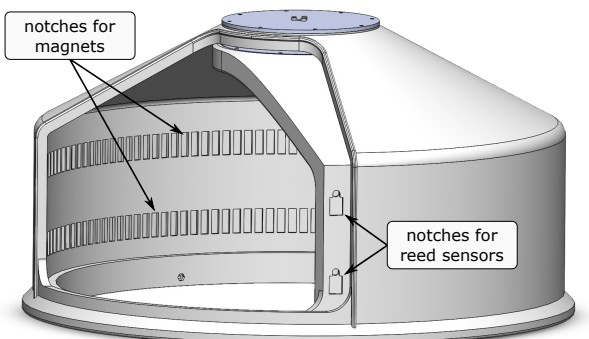

**Figure 6.** The image shows the cover with the notches for magnets and reed sensors to track the outer caps orientation.

To determine the outer cap's position, the caps provide notches for magnets and sensors as can be seen in Figure 6. Magnetic sensors at the inner cap combined with magnets at the outer cap are used to measure the position of the outer cap. The sensors are simple reed switches with an electrical contact that closes if the sensor is exposed to a magnetic field. One sensor is positioned in the upper notch to detect the closed position. A singe magnet in one of the notches opposed to the sensor activates

5   the reed contact, when the cover is closed.

To determine the distance and direction of any movement, two more sensors are installed next to each other in the lower sensor notch. Whenever a magnet passes by close to the sensors, one sensor will act a little earlier than the other. This delayed operation can be used to determine the direction of movement. Multiple magnets are distributed with alternating polarity around the outer cap opposed to the two sensors. Alternating the magnetic polarity creates a very weak field in between of two magnets

10   as shown in Figure 7b. Compared to that, if the magnets were placed with common polarity the magnetic field strength would



not vary very much along the magnets as depicted in Figure 7a. Since the reed switches are sensitive to the magnetic field strength and not to the polarity, this alternating arrangement ensures a much more reliable operation of the sensors.

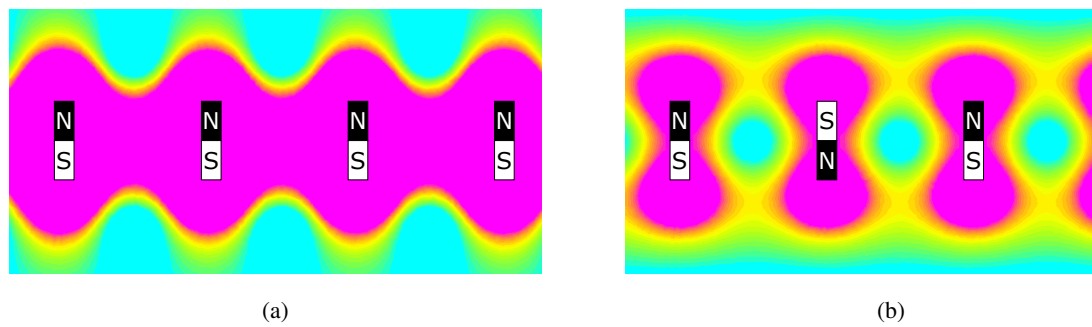

(a)                                                    (b)

**Figure 7.** Comparison of the magnetic field strength when placing the magnets in common polarity (a) and alternating polarity (b). Cyan represents a weaker magnetic field, whereas pink represent a stronger field.

To ensure that the sun path to the tracking mirrors will never be blocked, their azimuthal orientation is read directly from the mirror control. The corresponding position for the cover is determined and set as target position for the cover's control unit.

## 2.3 Thermal Regulation

There are several thermal concerns regarding the *EM27/SUN*. It is known, that the sun trackers stepping motors may block under freezing temperatures. A reason for this malfunction could be greasing. Another point is the unknown impact of heat on the system. Inside the *EM27/SUN* a InGaAs-detector senses the intensity of the NIR-light coming out of the interferometer. Typical InGaAs-detectors show a temperature-dependent transfer function (K.K. (2015); LLC (2004)). This leads to the desire of a constant operating temperature.

Besides these arguments, there is an even more critical point for a regulated temperature. In a situation, where the surrounding air warms up from a low temperature, for example in the morning, there is a high probability of condensation. In the case of warm air on cold equipment, water vapor in the air will condensate on the cold surfaces of the equipment and the tracking mirrors. This will distort the measured data or may even damage the electronics, and can potentially cause mirror degradations.

During the design, multiple options were discussed. A ventilation system combined with an electric heater was thought of, which has several downsides. First of all, high humidity may be vented into the enclosure, and therefore an air dehumidification system may be needed, too. Furthermore, a lot of openings would be necessary for the ventilation. This offers a high risk of leaks for rainwater and small animals to enter the enclosure. At last, a lot of heat may escape driven by wind blowing through the ventilation openings when heating is needed.

Thus another much easier approach that even offers cooling below the outside temperature was chosen. A thermo electric cooler (TEC) is installed in one of the walls of the base cabinet. The TEC-element is controlled by electrical current and transfers pure heat energy into or out of the enclosure. It does not require holes for ventilation and therefore keeps the enclosure




waterproof as well as locked for animals. Further, by controlling the temperature inside the enclosure with a target between 24 and 25°C, condensation is avoided.

To prevent unintended heat exchange between the inside and the outside of the enclosure a thermal insulation is installed. A 13 mm thick layer of special foam with a very low thermal conductivity, less than $0.040\,\mathrm{Wm^{-1}K^{-1}}$, covers the inside of the

whole base cabinet. This allows a low heat transport even on high thermal gradients between the inside and the outside.

| Component | max. Power |
|---|---|
| Spectrometer with solar-tracker | 45 W |
| Laptop | 30 W |
| Power supply and UPS | 35 W |
| Router, Voltage converters, etc. | $\sim 10\,\mathrm{W}$ |
| Total | $\sim 120\,\mathrm{W}$ |

**Table 1.** The table shows the expected maximum average power dissipation of all components inside the enclosure.

Because there are many heat sources inside the enclosure, cooling will be the more challenging than heating. The needed power for cooling is calculated by the given values of the component's specifications in their data sheets. A maximum average power of about 120 W (see Table 1) needs to be transfered out of the enclosure. The specification of the selected TEC gives a cooling capacity of 135 W. Thus about 15 W of extra cooling power will be available. Equation 1 is a rough estimate of the

thermal diffusion. It shows that the cooling system offers enough performance to cool the inside to about 2 K below the outside temperature.

$$\Delta T = \frac{\dot{Q}l}{\lambda A} = \frac{15\mathrm{W} \cdot 13\mathrm{mm}}{0.040\mathrm{Wm^{-1}K^{-1}} \cdot 2.48\mathrm{m^2}} \approx 1.97\ \mathrm{K} \tag{1}$$

where $\dot{Q}$ stands for the thermal power to transport along the distance $l$. $A$ is the area of the thermal conduction and $\lambda$ is the thermal conductivity of the material within the volume $A \cdot l$.

## 2.4   Rain Sensor

A *Hydreon RG-11* optical rain sensor is installed on top of the enclosure next to the cover. If rain is detected, the integrated logic of the enclosure will automatically close the cover and send a message to the measurement system. This message may be used for consecutive actions such as stopping ongoing measurements or informing an administrator.

## 2.5   Gas Spring

As described in Section 2.1 the enclosure's design is based on a control cabinet. In an upright position the cabinets door can easily be accessed and will stay in every position the user leaves it. Hence, it is not equipped with any door openers or springs.



In this application the cabinets orientation is changed, so that the door is now located on top. Therefore, the door's open positions are no longer stable and gravity will always push the door down. Hence, a gas spring is added to the door.

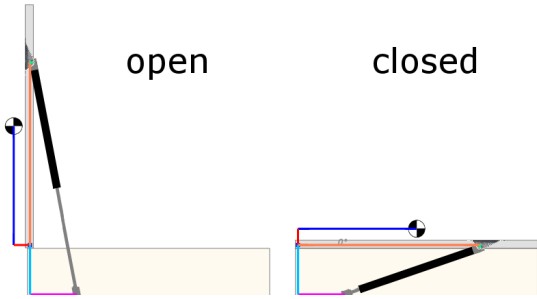

**Figure 8.** Simplified model of the enclosure and its door to calculate the joint positions and the strength of the gas spring.

A very simplified model as shown in Figure 8 is used for a proper positioning and dimensioning of the gas spring. It shows the geometries of joints as well as the door's point of gravity. The gas spring is calculated to not close by its own in any position. Thus, an operator can never be injured by the dropping door. The damping of the gas spring is very strong, so that any movement is curbed to very slow speed. Additional damping in the end positions ensures extra low accelerations, so that the cover will not be charged with strong impacts at the end positions.

The utilized gas spring has a length of 582 mm and a maximum travel of 250 mm. Its pushing force is about 180 N, which is necessary to lift the 15 kg of the door. The maximum force at the joints is calculated to 270 N.

## 2.6 Uninterruptible power supply

As already mentioned, the outer cap of the cover is driven by an electric motor. There is no mechanical fallback that may close the cover in case of a power outage. Hence, an uninterruptible power supply (UPS) is mandatory. Besides that, temporary power outages will destroy measurements and may place the system in an undefined state. Therefore, the UPS is designed powerful enough to supply the whole measurement system for several minutes.

## 2.7 Relays

While operating the *EM27/SUN* in the past, occasionally unexpected errors occurred. Some of these errors can only be resolved by restarting the spectrometer or its camera. Thus, two relays are included so that a remote operator, or an controlling software, can switch off the power to the spectrometer as well as disconnect its USB-Camera from the laptop.

The first relay is mounted on the DIN-rail and connects the power to the *EM27/SUN*. Besides resolving errors, this switch allows to power the instrument off during the night and thus helps to save energy.

The second relay is located on an USB-intermediate-plug. It was developed to allow the disconnection of the solar-tracker's USB-camera. In the past it has showed that the only effective solution to recover from camera errors is to physically reconnect the camera. Therefore, the relay on the intermediate-plug simply disconnects the 5V wire of the USB-connection, which





simulates physical unplugging of the USB-connector. As long as the relay is closed, the USB-data-communication will not be affected, since the data lines of the USB-connection are left untouched. With the help of this little circuitry, the USB-camera can be reset by a remote operator or automatically by a software.

## 2.8 Enclosure Control Board

The control board of the enclosure handles low level access to the enclosure's hardware. It provides protective safety features like closing the cover whenever an error happens. Furthermore, it receives commands from the laptop and operates the hardware of the enclosure accordingly. The control board works independently, so that a very high level of fail safety can be guaranteed. Almost every signal on and off the board is designed for the best possible fault tolerance to ensure a reliable operation in every situation.

Therefore, the control board implements the most critical safety features and serves as a tool for the laptop to operate the enclosure. The central brain on the board is an *Atmel ATmega168* microcontroller. It is a very robust system on chip (SOC), which includes a small, 20MHz RISC[1] processor with integrated memory. Besides a crystal and some capacitors, no external hardware is required, which makes it very reliable. For the communication with the laptop a *FTDI FT232RL* USB to RS232 converter is placed on the board. More details on the operation of the Enclosure Control Board and the communication interface

are given in Section 3.1.

## 3 Software

### 3.1 Enclosure Control Board Software

The enclosure control board is a central component of the enclosure. The software for the microcontroller on that board implements the enclosure's basic logic. It drives and controls the electric motor of the cover and reads its position by evaluating

the signals of the sensors in the cover. Moreover, it receives the signals of the rain sensor and UPS and communicates with the measurement system via USB. Controlling some relays, the enclosure control board can even power or unpower the spectrometer inside the enclosure. Self-monitoring and a fail-safe circuit design guarantee high reliability and security. This ensures full control and best protection of the instrument inside.

#### 3.1.1 Structure of the Software

For the best possible exploitation of the limited hardware resources, the software is split into a set of high and low priority tasks. The most important high priority task is the control of the cover as explained in Section 3.1.2. This process will be executed inside an interrupt at a fixed period of 10ms independent of the processor's load. The low priority tasks are the less time critical, communication processes. Resetting the hardware watchdog is also classified as low priority.

---

[1]Reduced Instruction Set Computer





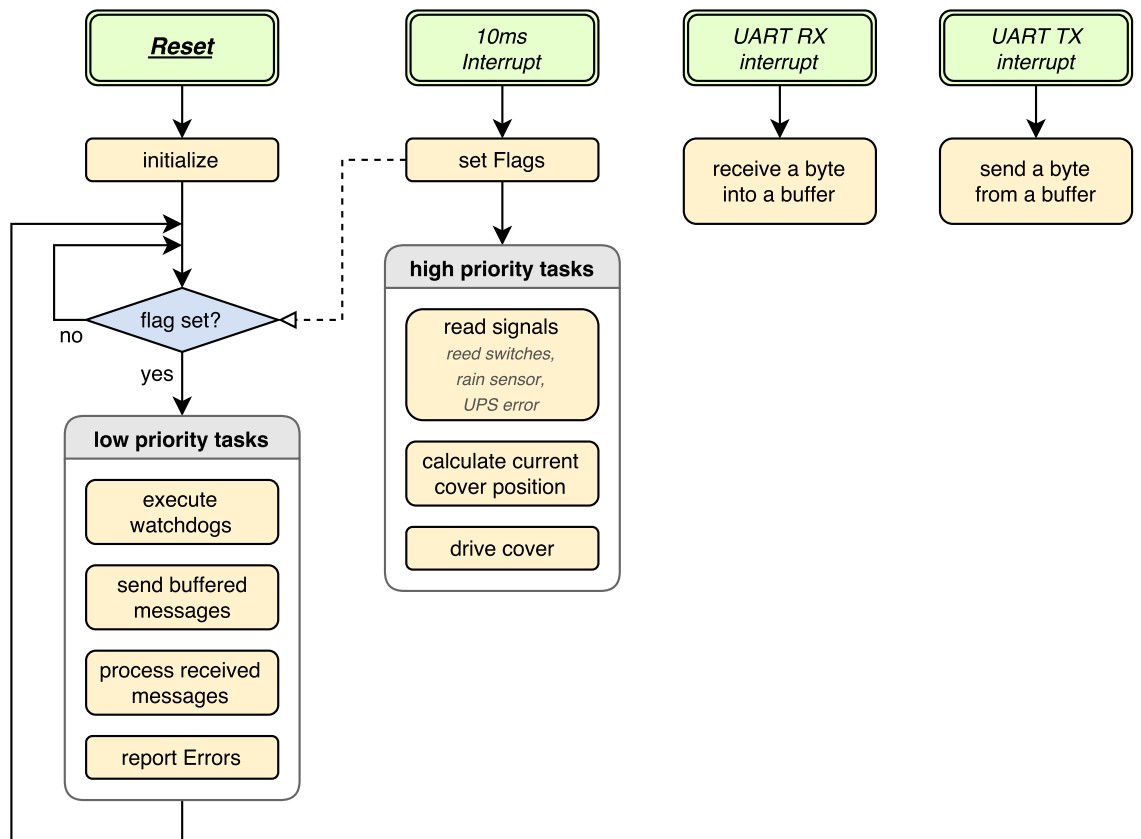

**Figure 9.** The flow chart visualizes the structure of the microcontroller's software.

A full overview of the program flow is shown in Figure 9. The reset node is the starting point of the program after every reset. After that, the processor initializes and enters its main loop, where it executes all the lower priority tasks. An interrupt will be executed every 10ms, processing the higher priority tasks. Additional to that, some flags will be set to synchronize the main loop to a more or less fixed execution interval.

5 **3.1.2 Cover Control Scheme**

Figure 10 depicts the control flow of the cover control. While operating in normal mode, the drive controller compares the current position to a target value. The target value could be derived from the azimuthal orientation for the tracking mirrors or the closed position. Depending on the distance between both, a target speed is calculated and fed into a ramp generator. The ramp generator slowly accelerates or decelerates the rotation of the cover to reduce mechanical stress. Its output signal
10 controls the motor driver, which includes the power amplifier to electrically drive the motor. The current position of the cover is calculated from the sensor readings and fed back to the drive controller. In case of any error, the target position will be set to the closed position, as explained before.





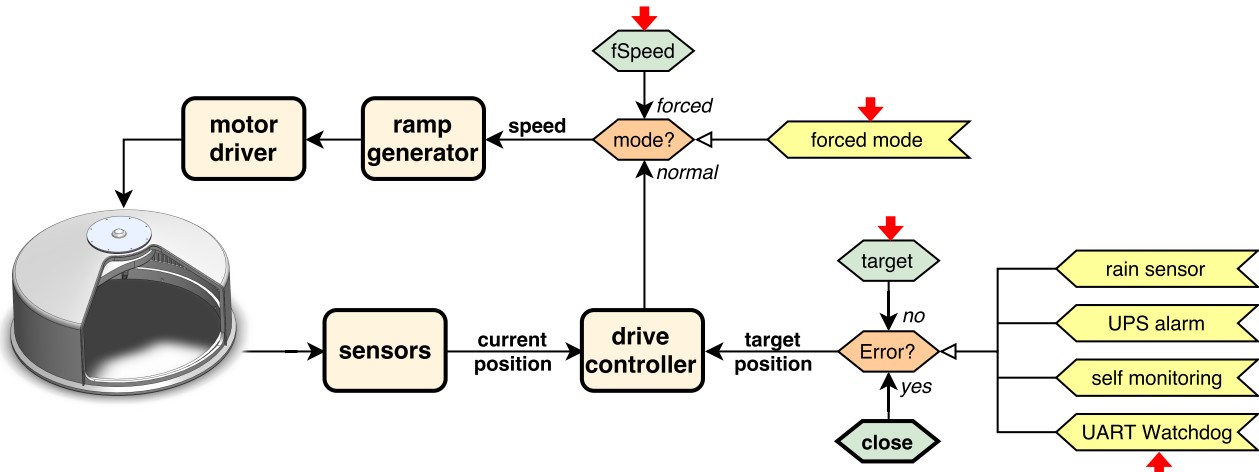

**Figure 10.** The flow chart shows how the control flow of the cover is designed. The red arrows indicate values that can be changed or influenced by the user via the USB-interface.

An additional forced mode was included to allow full control in case of an emergency situation. If any unexpected error happens, the loop can be cut open over the interface and a fixed value can be fed into the ramp generator. Hence, a remote operator can take full control over the cover and navigate the cover into a secure position if needed.

### 3.1.3 Safety Features

The microcontroller's hardware integrates a watchdog feature that monitors the processor's operation. A watchdog basically is a timer that expects a reset signal within its configurable period. In normal operation the watchdog timer will be reset before a timeout occurs. However, if the processor hangs at any position in the program the watchdog timer will time out and consequently trigger a function. In case of the hardware watchdog, a full reset will be performed. Thus, a very high reliability of operation is guaranteed.

Additionally, the software provides an optional UART watchdog. When enabled it will expect an alive message within every 5 seconds. In contrast to the hardware watchdog, which is not optional, a timeout of the UART watchdog will not trigger a system reset. Instead an error-flag will be set and the program will consequently close the cover as can be seen in Figure 10.

### 3.2 Computer Software

We also developed a computer software to offer a set of methods to control the enclosures functionalities with a graphical user
interface (GUI). The software is programed in Python. The GUI is called *ECon*, which is derived from "Enclosure Control" and appears as a window on the desktop (Figure 11).



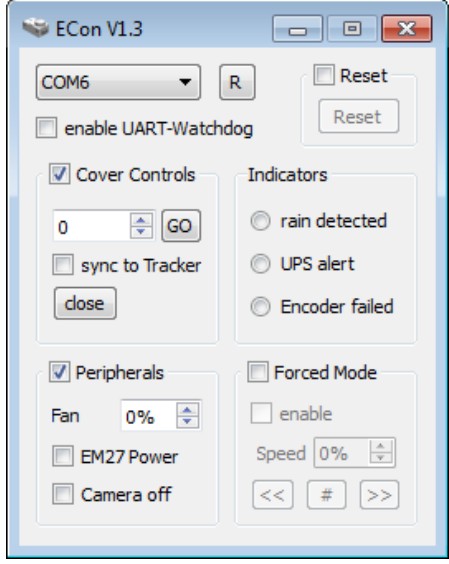

**Figure 11.** The image shows the graphical user interface which is called ECon. With the given controls and indicators nearly every function of the enclosure can be controlled and observed.

## 4    Results

After building the enclosure its functions were tested and proved. First of all, the dryness of the enclosure after extreme stormy and snowy condition is affirmed. Secondly, the ability to regulate the temperature inside the enclosure is examined. The capability to not block the sun during the course of the day is verified. Finally, the remote controllability and operability is shown.

### 4.1    Rain and Snow Test

Several extreme weather conditions acted on the enclosure. The surveillance camera, which has been installed to allow a live view on the enclosure from remote, captured that situation in Figure 12. Even after these extreme rain and wind conditions the enclosure was completely dry and not a single drop of water penetrated it. Figure 13 shows that no water was blown into the cover. The rain sensor also reliably detected the first drop of water on its sensitive area and the cover was closed within less than 6 seconds.

In winter 2016 Munich was experiencing a lot of snow, and the enclosure proved to be robust against snow fall on the cover (Figure 14). The enclosure is constantly heated and the spectrometer is able to carry out measurements whenever it is sunny, even if the ambient temperature falls below 0°C.





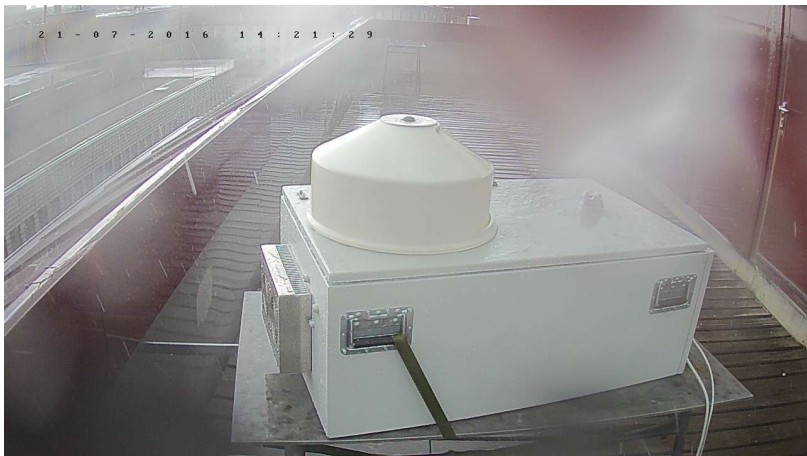

**Figure 12.** Impression of the bad weather the fully equipped enclosure was exposed to.

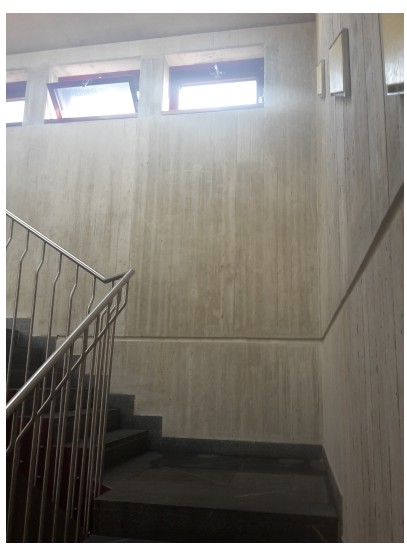
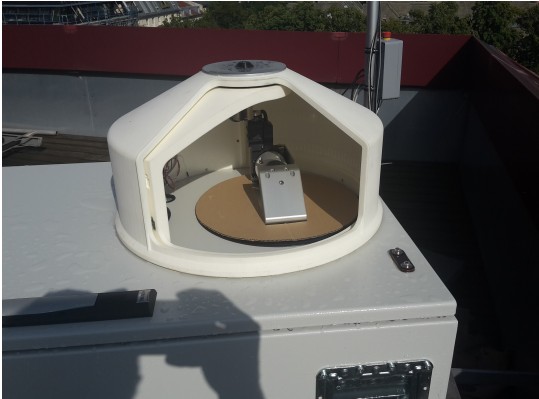

**Figure 13.** (a) Image of flooded stair house. The water was blown into the tilted window. (b) Image of the open enclosure after the storm. As can be seen, no water was able to penetrate the cover.

## 4.2 Unblocked Sunlight

During the one year test period, the enclosure has shown no single case of blocking the sun during measurements. The test azimuthal angle spans from 71.4° to about 288.5°, which corresponds to the range reachable on the longest day of the year (21 June) when the solar zenith angles are below 75° (see Figure 15).





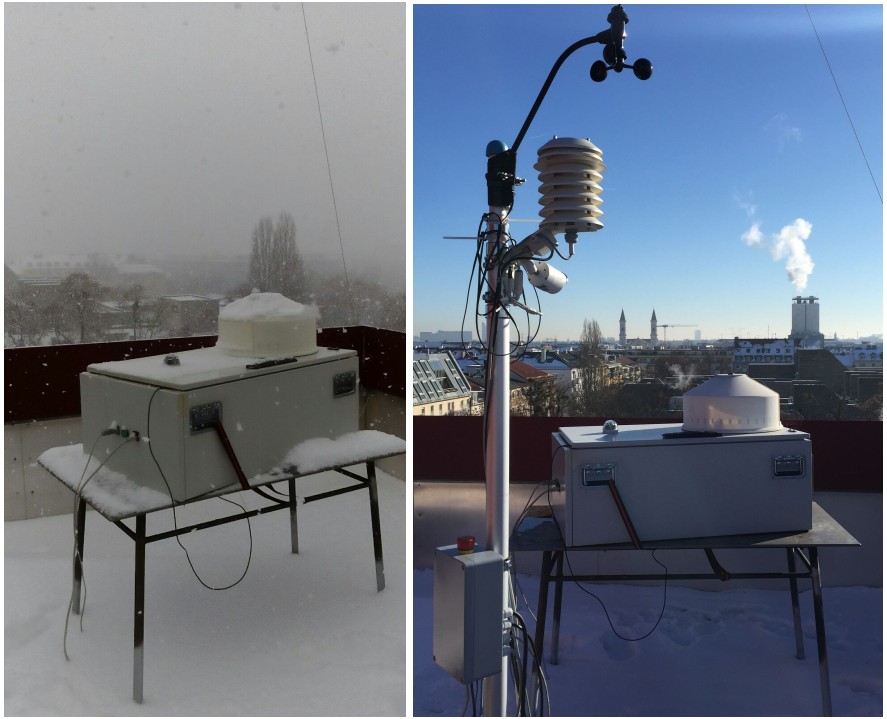

**Figure 14.** (a) During 2016 winter, inches of snow fall on the enclosure with the spectrometer located inside. (b) The system is taking measurements in cold winter below 0°C.

### 4.3 Thermal Regulation

The enclosure's thermal regulation was tested by logging thermal readings of the outside and inside temperatures during normal operations. Figure 16 shows the logged data. As can be seen, the internal temperature of the enclosure does not show a very smooth temperature curve. This is mainly because of the regulator, which follows a very simple three-point control scheme.

5 If the temperature falls below the configured threshold of 24°C the TEC is turned on for heating until the temperature raises above 24.1°C. The same is valid for cooling whereby the TEC is powered for cooling at above 25°C and disabled at 24.9°C. The regulator only has the three states heating, cooling and off. There is no intermediate state, which would allow for a more smooth regulation. Thanks to the temperature control, no condensation was observed on the mirrors or other components inside of the base cabinet.

10 While construction, the TEC was designed to pump out the heat of the measurement system. Cooling below the outside temperature were considered to be a useful feature after the enclosure was build up. The build-in reserves of the TEC are not large enough to cool the system more than just a few degree below the outside temperature, as can be seen in Figure 16. On 23rd of July the temperature can be controlled within the desired range as long as the outside temperature is lower than 27°C or the measurement system is turned off, such that less heat is generated.



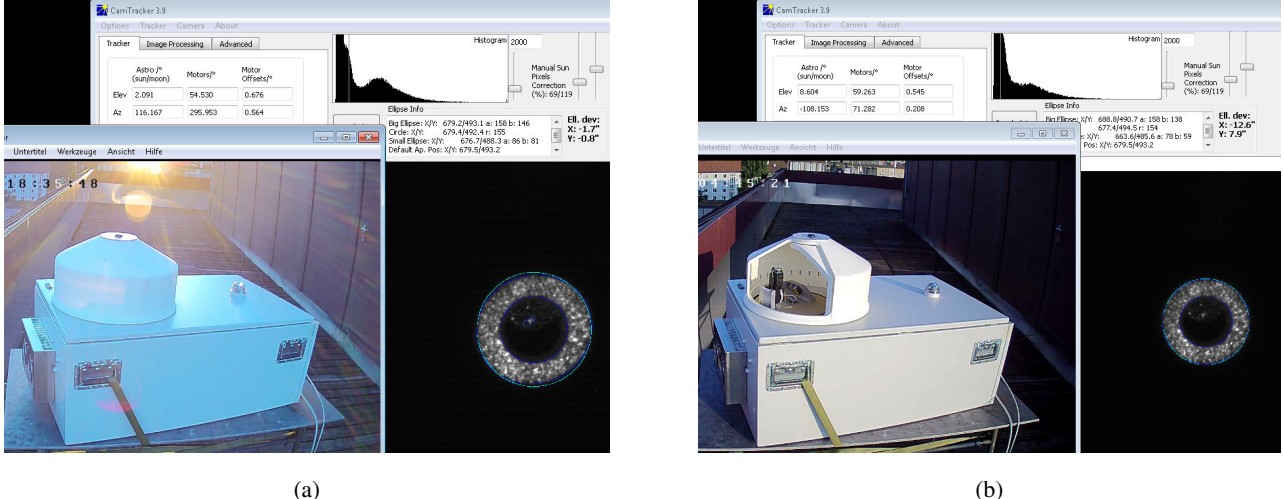

(a)                                          (b)

**Figure 15.** The images show screen shots of the measurements laptop in the late evening (top) and early morning (bottom). In the image on the left, one can see the instrument tracking the sun at an azimuthal angle of 296° on 29 July at 6:35pm. The right image shows the the tracker following the sun at about 71° azimuthal angle on 30 July at 4:45am. As expected, in both situations the cover did not block the sunlight.

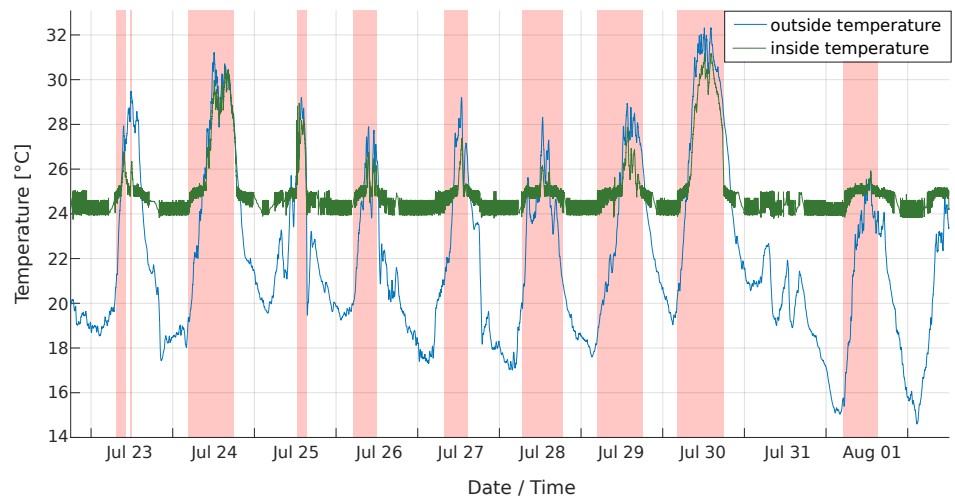

**Figure 16.** The diagram shows the internal and external temperature of the enclosure. The red bars illustrate time periods during which the measurement system was active and therefore produced heat that is needed to be pumped out of the enclosure by the TEC.





## 4.4 Remote Operability

The measurement system can be fully remote operated. Using a remote desktop software any operator can log in from any computer and take control over the system. Thanks to the enclosure there is no need for physical attendance to start, stop or monitor measurements.

5  Figure 17 shows the daily amount of collected data. As can be seen, the density increased clearly when the enclosure was set up for operation. Nearly every day measurements were taken, even small windows in the clouds were used to sample the data.

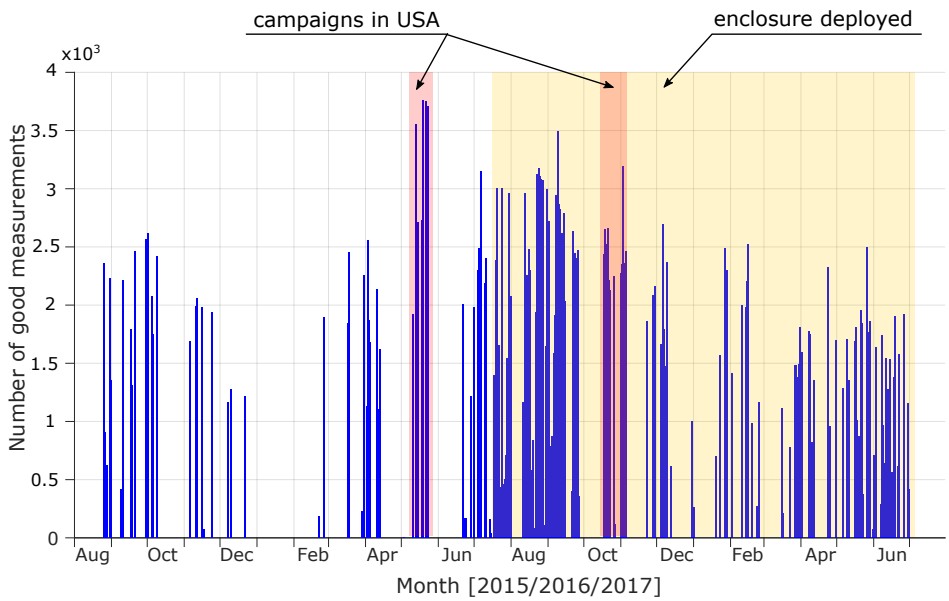

**Figure 17.** Visualization of the amount of daily sampled data with the *EM27/SUN*. The yellow region marks the time duration when the *EM27/SUN* is deployed inside of the enclosure. During time periods indicated by the red regions, the *EM27/SUN* was participating the measurement campaigns in USA, Indianapolis and San Francisco Bay Area, respectively. Due to the shipping time to and back from USA, the measurements were not taken a month before and after the campaigns.

## 5 Future Work

In future, rain prediction information can be integrated to assist the decision making. One potential approach could be to assess
10  real time on-line rain radar data and meteorological forecasts to predict the upcoming weather situations.

Further improvements of the enclosure could focus on a localized thermal regulation. One could specify the most sensitive components of the measurement system and give particular attention to their thermal regulation. This may also accomplish the thermal demands while reducing the total energy consumption.



Another enclosure improvement could target a lightweight and portable version, which would be better suited for mobile measurements as needed in campaigns. The current enclosure was designed for stationary use and gives proof of the concept. Nevertheless, there is a high demand to use the enclosure for campaigns. This raises a whole new set of requirements such as reduced weight, size and power consumption.

**6 Conclusion**

An automated enclosure for atmospheric solar-tracking instruments, in our case *EM27/SUNs*, was designed, engineered, assembled, and successfully tested under extreme weather conditions. The automated solar-tracking system is located in central Munich (48.15N, 11.57E).

It is suited for fully autonomous operation, and enables facile handling of the measurement system. When potentially harmful
weather conditions arise, the system can be shut down and protected within seconds. In case of unexpected rain, the enclosure reliably protects the measurement system by closing its cover. In this manner, the measurement system is sheltered from heavy storms, rainfalls, and snowfalls without the need for any physical human interaction.

Remote access can be gained by any smartphone or computer. Thus, the system can be observed and controlled within a few seconds from anywhere in the world. This lowers the inhibition level to start measurements immensely, resulting in a
significantly increasedamount of collected data. Furthermore, the automated enclosure reduces the need for costly manpower. This leads to a much higher efficiency of the measurement system. The increased amount of collected data will finally optimize the chance to sample good data even during periods of instable weather conditions. This will increase the significance of any scientific outcome that is derived from these data.

All in all, the enclosure provides the main functionality for a complete automation of the measurement system, it guarantees
maximization of collected data amount while minimizing the operational risks and costs, and thus provides a fundament for a long-term GHG monitoring sensor network.

Acknowledgments. We gratefully thank Gerold Wunsch for helping with the cover concept development. We also acknowledge our Bruker colleagues Peter Maas, Gregor Surawicz for technical support. Further, we thank Bruce Daube, Steven C. Wofsy, Andreas Meichelböck, Johannes C. Paetzold, Duc Hai Nguyen, and Patrick Aigner for fruitful discussions.

Jia Chen and Ludwig Heinle are supported by Technische Universität München - Institute for Advanced Study, funded by the German Excellence Initiative and the European Union Seventh Framework Programme under grant agreement n° 291763.





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
