# Peer review of "Automated Enclosure and Protection System for Compact Solar-Tracking Spectrometers"

_Atmospheric Measurement Techniques, 2017_

## Referee Comment (RC1) · Anonymous Referee #1 · 16 Sep 2017

This manuscript clearly and exhaustively describes an enclosure for a mobile solar-tracking spectrometer that can allow for greater measurement frequency and improved protection for the instrument against rain or other bad weather. The manuscript is very readable and describes the system effectively, but could use a small bit of English usage editing. There are a few small points that I see as useful additions:

Size, power, and weight are not described.

The measurements use a laptop computer, which is not shown in the schematic diagrams. While it is clear that most operations will use remote access software to control that laptop computer, it may prove necessary to access the computer occasionally. Does physical access to the computer require opening the enclosure?

[Figure]

The cover rotation encoder appears to be a relative encoder, gaining absolute position information by seeking a limit switch, but the text is not fully clear on this point. Please give a little more detail.

Minor / typographical points:

Page 1, line 4: The measurements are are not truly continuous in time; they require direct solar viewing (clear skies, daytime). Maybe find a better wording?

page 1, line 11: I think the authors mean "foundation" instead of "fundament"

page 6, section 2.2.2: This is a good calculation of the azimuthal dependence of the viewing geometry, but it appears that the scanner cannot view the zenith, so at some low latitudes where the sun can get close to the zenith, the cover will block the view, causing a gap near local solar noon. Please provide a calculation of this effect.

page 7, line 4: I think you mean "single" magnet where it says "singe"

Page 9, line 2: It appears that there is no seal between the circular tracker base ring and the enclosure, so it seems like ambient air (and humidity) can get into the enclosure. If the ambient T > 25°C and RH is high, condensation is possible. Please describe if the system is sealed at this point or not and if so how.

Table 1 shows a laptop in the power budget, but the system drawing (Figure 2) does not show the laptop.

page 12, section 3.1.2: It appears that the encoder for the motion of the moving cover determines relative motion direction and "steps" as each magnet is passed. This type of controller needs a limit switch to determine absolute position, which is presumably the closed switch. Therefore, the microcontroller needs to keep track of the absolute position in software. Potentially this is not fully explained, or potentially there is some absolute position encoding with the magnet scheme that needs further explanation.

page 19, top: A lighter weight version of the enclosure is mentioned, but I don't think

that the actual weight of the current system is discussed. It would be valuable to give the size and weight of the current enclosure system.

page 19, line 15: missing a space in the two words "increased amount"

page 19, line 20: Again, I think "fundament" should be "foundation"

---

## Referee Comment (RC2) · Anonymous Referee #2 · 25 Oct 2017

The authors successfully describe in considerable detail, their design for an automated enclosure and protection system suitable for the Bruker EM27/SUN FTS. There are possibly as many solar-tracking enclosure designs as there are solar tracker designs – as these all tend to be somewhat custom in nature. In this instance, it is likely the authors' version may prove especially interesting because the EM27/SUN is proving a popular instrument, well-suited to outdoor autonomous operation, and would benefit from a successful design as described here. The paper is well-written and interesting, but would benefit from further editing for minor English grammar corrections to aid the reading flow, and to shorten the overall document. The drawings are excellent.

Specific comments

Page 1: Line 8: The words "less than" are redundant and can be deleted, as this

meaning is covered by "within". Line 10: "fundament" is not in common usage, I suggest replacing with "basis". Line 15: "wellbeing" should be "well-being".

Page 2: Line 1: "weight" should be "weigh".

In the remainder of page 2, I feel that the discussion on the merits of the 125HR could be shortened or omitted. For example, the detail concerning "scanner wear" is not at all relevant to the paper, nor to the choice between using a 125HR or EM27/SUN. It would suffice to state, in one paragraph only, the advantages of using the EM27/SUN (size, portability, rapid-deployment, nature of measurements capability etc.) and hence the need for an excellent enclosure and protection system.

Page 3: This page repeats or builds-upon much of what was discussed in page 2, i.e. water/weather-proof requirement and human effort etc. Consider stating these requirements concisely only once.

Page 6: Line 8: Are measurements not made for SZA greater than 80 deg? This might be somewhat limiting the application of the cover for other uses. The reviewer often begins measurements at 88 SZA. Please clarify if the cover can be used to the horizon (90 SZA).

Page 7 (and 6): No mention is made of the potential for any trapped water to freeze, jamming the cover movement. Has this occurred or likely to? If not, then consider mentioning steps taken to prevent this happening.

Page 8: Line 6: "block" should be "seize" or "stall".

In 2.3, Thermal Regulation, again there may be a bit too much detail (e.g. mention of InGaAs detector…). Consider reducing the discussion in the first 3 paragraphs to perhaps a single paragraph that justifies the excellent decision to use a TEC unit for thermal control. Consider too explaining that the TEC uses solid-state Peltier devices (some readers may be more familiar with this name).

Page 9: There is good detail concerning the thermal calculations. I would like to see

another sentence or two explaining why the RG-11 rain sensor was chosen – I did research this unit and it sounds an excellent choice.

Page 10: Line 7: "charged with" could be replaced with "subject to".

Concerning the UPS, it would be useful if a power outage could initiate a controlled shut down of the instrument and computer. Is this done?

Page 11: Line 13: "a FTDI..." should be "an FTDI…" (F has a vowel sound "eff").

Page 14: Figure 11 caption: a coma is needed after "indicators". Line 3: "affirmed", more common usage would be "confirmed". Line 12: "snow fall" should be "snowfall".

Page 16: Figure 14 caption "snow fall" to "snowfall". 4.3, Thermal regulation: The reviewer has used similar TECs and had much success replacing the simple thermostat with an electronic PID controller, with parameters achieved using a self-learn function.

Page 19: Line 9: "facile" would be more commonly replaced by "easy" in this use. Line 15: "manpower" could be made gender-neutral by using "labour" or "human effort" instead.

Well done on the excellent design of your unit!

---

## Author Comment (AC1) · 4 Feb 2018

We would like to thank the reviewers for carefully reading the paper and giving helpful comments. Below, the reviewers' original text is included. The answers are highlighted in blue.

**Review** **1:**
**General comments:**

This manuscript clearly and exhaustively describes an enclosure for a mobile solar tracking spectrometer that can allow for greater measurement frequency and improved protection for the instrument against rain or other bad weather. The manuscript is very readable and describes the system effectively, but could use a small bit of English usage editing.

Thank you. To improve the English writing, we have carefully gone through the paper several times and consulted with a professional English Coach. We have corrected grammar mistakes and reduced redundancy in the paper. We will also use the English copy-editing services from Copernicus after the paper is accepted.

There are a few small points that I see as useful additions:

1) Size, power, and weight are not described.

Thank you for this comment. In Table 2 we have already listed the power consumption of the system components, and the total power consumption of the sensor system including the enclosure, which is about 120 W. The size of the base cabinet is 112x62x41 cm, and the weight of the enclosure including spectrometer is around 100 kg. We added Table 1 for listing the dimension and weight information.

2) The measurements use a laptop computer, which is not shown in the schematic diagrams. While it is clear that most operations will use remote access software to control that laptop computer, it may prove necessary to access the computer occasionally. Does physical access to the computer require opening the enclosure?

Yes, everything is inside the enclosure to be protected against harmful weather conditions. Thus, opening the enclosure will be required for accessing the components in case of malfunctions or maintenance. The computer does not necessarily have to be a laptop computer, it can be any kind of computer, running Windows, which is required for the measurement software of the instrument.

3) The cover rotation encoder appears to be a relative encoder, gaining absolute position information by seeking a limit switch, but the text is not fully clear on this point. Please give a little more detail.

Thank you for this note. We have modified the text in Section 2.2.4 to be more precise:

"To determine the outer cover's position, we have developed an encoder using magnets and reed sensors that are placed inside of the notches on the covers (Figure 6). With this concept, the absolute cover position can be determined and the spectrometer is reliably protected.

Three reed sensors are glued in the notches of the inner cover, and each of the sensors has an electrical contact that closes if the sensor is exposed to a magnetic field. Nineteen neodymium magnets with a magnetic flux of 1.17 Tesla each were distributed in the notches of the outer cover. One reed sensor and a single magnet positioned in the upper notches are used to detect the closed position (absolute zero position). From then on, the motion direction is determined and the number of magnets are counted. The position can be determined with a precision of about 10 degrees. "

**Minor / typographical points:**

Page 1, line 4: The measurements are not truly continuous in time; they require direct solar viewing (clear skies, daytime). Maybe find a better wording?

Thank you. We changed the sentence to:

"It has provided ground-based measurements of column-averaged concentrations of $CO_2$, $CH_4$, $O_2$, and $H_2O$ throughout this time."

page 1, line 11: I think the authors mean "foundation" instead of "fundament"

Thank you, we have corrected this.

page 6, section 2.2.2: This is a good calculation of the azimuthal dependence of the viewing geometry, but it appears that the scanner cannot view the zenith, so at some low latitudes where the sun can get close to the zenith, the cover will block the view, causing a gap near local solar noon. Please provide a calculation of this effect.

Thanks for this comment. However, this calculation is really complex due to the following factors:

1. First, the mirror is not at the azimuthal rotational axis.
2. The azimuthal rotational axis is not concentric with the cover rotational axis
3. The elevation limits are dependent on the azimuthal position
4. The minimum exposure surface of the mirrors need to be defined

Nevertheless, we made an estimation using the 3D CAD model and added the sentence "A rough estimation based on the 3D CAD model shows that the permissible solar zenith angle range is 23(+/-1) to 88(+/-1) degrees." to section 2.2.2.

page 7, line 4: I think you mean "single" magnet where it says "singe"

Yes, we have corrected this typo.

Page 9, line 2: It appears that there is no seal between the circular tracker base ring and the enclosure, so it seems like ambient air (and humidity) can get into the enclosure. If the ambient T > 25C and RH is high, condensation is possible. Please describe if the system is sealed at this point or not and if so how.

There is a 2 cm spacing between the circular tracker base plate and the covers. There is no sealing, probably not even wanted, since the tracker itself provides an open path into the base cabinet (where the light beam goes). In the current construction it is most likely, that the air passes by the trackers mirrors and hit onto the cold surface of the TEC. We never experienced condensation on the TEC, which is much cooler than the Instrument. The instrument is permanently heated by itself.

Table 1 shows a laptop in the power budget, but the system drawing (Figure 2) does not show the laptop.

The Laptop is placed on top of the EM27/SUN spectrometer to ensure best accessibility. We added this information to the caption of Figure 2.

page 12, section 3.1.2: It appears that the encoder for the motion of the moving cover determines relative motion direction and "steps" as each magnet is passed. This type of controller needs a limit switch to determine absolute position, which is presumably the closed switch. Therefore, the microcontroller needs to keep track of the absolute position in software. Potentially this is not fully explained, or potentially there is some absolute position encoding with the magnet scheme that needs further explanation.

Yes, you are right. The absolute zero position is provided when the magnet actives the reed contact in the upper notch. From then on, the motion direction is determined and the number of magnets are counted.

We added this information to section 2.2.4

page 19, top: A lighter weight version of the enclosure is mentioned, but I don't think that the actual weight of the current system is discussed. It would be valuable to give the size and weight of the current enclosure system.

We included the size and weight of the enclosure system in Table 1.

page 19, line 15: missing a space in the two words "increased amount"

We have modified it, thank you.

page 19, line 20: Again, I think "fundament" should be "foundation"

We have modified it, thank you.

**Review 2:**

The authors successfully describe in considerable detail, their design for an automated enclosure and protection system suitable for the Bruker EM27/SUN FTS. There are possibly as many solar-tracking enclosure designs as there are solar tracker designs – as these all tend to be somewhat custom in nature. In this instance, it is likely the authors' version may prove especially interesting because the EM27/SUN is proving a popular instrument, well-suited to outdoor autonomous operation, and would benefit from a successful design as described here. The paper is well-written and interesting, but would benefit from further editing for minor English grammar corrections to aid the reading flow, and to shorten the overall document. The drawings are excellent.

Thank you. To improve the English writing, we have carefully gone through the paper several times and consulted with a professional English Coach. We have corrected grammar mistakes and reduced redundancy in the paper. We will also use the English copy-editing services from Copernicus after the paper is accepted.

**Specific comments:**

Page 1: Line 8: The words "less than" are redundant and can be deleted, as this meaning is covered by "within". Line 10: "fundament" is not in common usage, I suggest replacing with "basis". Line 15: "wellbeing" should be "well-being". Page 2: Line 1: "weight" should be "weigh".

Thank you, we have modified these points.

In the remainder of page 2, I feel that the discussion on the merits of the 125HR could be shortened or omitted. For example, the detail concerning "scanner wear" is not at all relevant to the paper, nor

to the choice between using a 125HR or EM27/SUN. It would suffice to state, in one paragraph only, the advantages of using the EM27/SUN (size, portability, rapid-deployment, nature of measurements capability etc.) and hence the need for an excellent enclosure and protection system.

Thank you for the suggestion. We have shortened the paragraph regarding 125HR, but not omitted it completely. We think that some readers would appreciate the information to understand the benefits. We would like to keep the information. People who know, can just quickly read over it, whereas others will be happy about the information.

Page 3: This page repeats or builds-upon much of what was discussed in page 2, i.e. water/weather-proof requirement and human effort etc. Consider stating these requirements concisely only once.

We have reduced the redundant information and stated the requirements only once in the introduction.

Page 6: Line 8: Are measurements not made for SZA greater than 80 deg? This might be somewhat limiting the application of the cover for other uses. The reviewer often begins measurements at 88 SZA. Please clarify if the cover can be used to the horizon (90 SZA).

We added the sentence "A rough estimation based on the 3D CAD model shows that the permissible solar zenith angle range is 23(+/-1) to 88(+/-1) degrees." to section 2.2.2.

Page 7 (and 6): No mention is made of the potential for any trapped water to freeze, jamming the cover movement. Has this occurred or likely to? If not, then consider mentioning steps taken to prevent this happening.

No, this has not been occurred and is also not likely. The enclosure is heated to 25 degrees. The warm air rises up and prevent freezing of trapped water. We have added this comment to section 4.1.

Page 8: Line 6: "block" should be "seize" or "stall".

We changed it to "stall", thank you.

In 2.3, Thermal Regulation, again there may be a bit too much detail (e.g. mention of InGaAs detector: : :). Consider reducing the discussion in the first 3 paragraphs to perhaps a single paragraph that justifies the excellent decision to use a TEC unit for thermal control. Consider too explaining that the TEC uses solid-state Peltier devices (some readers may be more familiar with this name).

We have reduced the discussion in section 2.3, and compressed the three paragraphs to one. Also we indicate TEC uses solid-state Peltier devices.

Page 9: There is good detail concerning the thermal calculations. I would like to see another sentence or two explaining why the RG-11 rain sensor was chosen – I did research this unit and it sounds an excellent choice.

We added the following paragraph:

"The sensor works as follows: an infrared LED emits light into a dome-shaped lens. The light is reflected by the lens surface and travels along its shape to the other side where it strikes a photodiode, which monitors the light intensity. When a water drop lands on the surface of the lens, less light will be detected by the photodiode because the drop allows the light to escape by refraction. RG-11 was chosen due to its dome-shaped surface. It is an improvement over a flat surface as the drops do not accumulate on the surface, which could produce a false signal after the rain stops."

Page 10: Line 7: "charged with" could be replaced with "subject to".

Done

Concerning the UPS, it would be useful if a power outage could initiate a controlled shut down of the instrument and computer. Is this done?

No currently it is not. However, we could implement the controlled shut down of the computer in the future.

Page 11: Line 13: "a FTDI…" should be "an FTDI: : :" (F has a vowel sound "eff").

Done

Page 14: Figure 11 caption: a coma is needed after "indicators". Line 3: "affirmed", more common usage would be "confirmed". Line 12: "snow fall" should be "snowfall".

Page 16: Figure 14 caption "snow fall" to "snowfall".

Thank you, we have improved these language items according to your suggestions.

4.3, Thermal regulation: The reviewer has used similar TECs and had much success replacing the simple thermostat with an electronic PID controller, with parameters achieved using a self-learn function.

Thanks for the suggestion! We could think about something similar in future.

Page 19: Line 9: "facile" would be more commonly replaced by "easy" in this use.

We changed it.

Line 15: "manpower" could be made gender-neutral by using "labour" or "human effort" instead.

Done

Well done on the excellent design of your unit!

Thank you very much for your compliment and recognition!

[revised manuscript text omitted]